# Report

EMBO
Molecular Medicine

# Polysialic acid blocks mononuclear phagocyte reactivity, inhibits complement activation, and protects from vascular damage in the retina

Marcus Karlstetter[1,2], Jens Kopatz[3], Alexander Aslanidis[1], Anahita Shahraz[3], Albert Caramoy[1], Bettina Linnartz-Gerlach[3], Yuchen Lin[4], Anika Lückoff[1], Sascha Fauser[1], Katharina Düker[3], Janine Claude[3], Yiner Wang[3], Johannes Ackermann[3], Tobias Schmidt[5], Veit Hornung[5,6], Christine Skerka[4], Thomas Langmann[1,\*,†] (ID) & Harald Neumann[3,\*\*,†] (ID)

## Abstract

**Age-related macular degeneration (AMD) is a major cause of blindness in the elderly population. Its pathophysiology is linked to reactive oxygen species (ROS) and activation of the complement system. Sialic acid polymers prevent ROS production of human mononuclear phagocytes via the inhibitory sialic acid-binding immunoglobulin-like lectin-11 (SIGLEC11) receptor. Here, we show that low-dose intravitreal injection of low molecular weight polysialic acid with average degree of polymerization 20 (polySia avDP20) in humanized transgenic mice expressing SIGLEC11 on mononuclear phagocytes reduced their reactivity and vascular leakage induced by laser coagulation. Furthermore, polySia avDP20 prevented deposition of the membrane attack complex in both SIGLEC11 transgenic and wild-type animals. *In vitro*, polySia avDP20 showed two independent, but synergistic effects on the innate immune system. First, polySia avDP20 prevented tumor necrosis factor-α, vascular endothelial growth factor A, and superoxide production by SIGLEC11-positive phagocytes. Second, polySia avDP20 directly interfered with complement activation. Our data provide evidence that polySia avDP20 ameliorates laser-induced damage in the retina and thus is a promising candidate to prevent AMD-related inflammation and angiogenesis.**

**Keywords** age-related macular degeneration; complement; microglia polysialic acid; SIGLEC

**Subject Categories** Immunology; Neuroscience; Pharmacology & Drug Discovery

## Introduction

Age-related macular degeneration (AMD) is the main cause for visual impairment and legal blindness in the industrialized world as more than one-third of the population over the age of 75 develops AMD (Augood *et al*, 2006). AMD is associated with chronic innate immune activation specifically involving the complement system, activation of retinal phagocytes, and production of reactive oxygen species (ROS) (Ambati & Fowler, 2012). Late-stage disease can manifest as either geographic atrophy or choroidal neovascularization. The latter form is treated with intravitreally injected drugs targeting vascular endothelial growth factor (VEGF) (Rofagha *et al*, 2013). There is currently no approved drug treatment for dry AMD/geographic atrophy.

Genome-wide association studies have clearly shown that genetic variants regulating the complement system are associated with AMD (Fritsche *et al*, 2014, 2016). The alternative complement pathway is over-activated in AMD and insufficiently controlled in patients with AMD-associated polymorphisms of complement factor H (CFH) and complement factor I (CFI) (Bradley *et al*, 2011).

Highly reactive mononuclear phagocytes were found in the outer retina of AMD patients with geographic atrophy (Gupta *et al*, 2003) and in the mouse model of laser-induced choroidal neovascularization (Lückoff *et al*, 2016), but their contribution to disease progression is still unclear. These reactive phagocytes contain metabolite particles and are tightly associated with drusen (Killingsworth *et al*, 1990; Mullins *et al*, 2000). NADPH oxidase of reactive phagocytes is the principal source of overt neurotoxic amounts of ROS (Gao *et al*, 2012; Claude *et al*, 2013; Bodea *et al*, 2014). Thus, phagocytes could act beneficial by clearing tissue debris, but could also have a

1 Laboratory for Experimental Immunology of the Eye, Department of Ophthalmology, University of Cologne, Cologne, Germany
2 Therapeutic Research Group Ophthalmology, Bayer Pharma AG, Wuppertal, Germany
3 Institute of Reconstructive Neurobiology, University Hospital Bonn, University of Bonn, Bonn, Germany
4 Leibniz Institute for Natural Product Research and Infection Biology, Jena, Germany
5 Institute of Molecular Medicine, University Hospital Bonn, University of Bonn, Bonn, Germany
6 Gene Center and Department of Biochemistry, Ludwig-Maximilians-Universität München, Munich, Germany
  \*Corresponding author. Tel: +49 221 478 7324; E-mail: thomas.langmann@uk-koeln.de
  \*\*Corresponding author. Tel: +49 228 6885 541; E-mail: hneuman1@uni-bonn.de
  †These authors contributed equally to this work

neurotoxic potential by production of ROS. Therefore, selective modulation of phagocyte neurotoxicity should be considered as a treatment for immune-associated retinal diseases.

Polysialic acid (polySia), an extended homopolymer of α2.8-linked sialic acids, is linked to mammalian glycoproteins (NCAM, synCAM-1, neuropilin-2, CD36) on the cell surface of neurons and immune cells (Sato & Kitajima, 2013). Besides involvement of poly-Sias in synaptic development and wiring of retinal ganglion cell axons, sialic acids as the terminal caps of the sugar branches play an essential role in self-recognition and inhibition of the innate immune system (Monnier *et al*, 2001; Varki, 2011; Hildebrandt & Dityatev, 2015). Sialic acids were shown to modulate alternative complement processes in a CFH-dependent manner (Blaum *et al*, 2015). Furthermore, neuronal sialic acids and polySia with an average degree of polymerization of 20 (avDP20) are recognized by the immunoreceptor tyrosine-based inhibitory motif (ITIM)-bearing sialic acid-binding immunoglobulin-like lectin-11 (SIGLEC11) receptor and reduce inflammatory neurotoxicity of phagocytes in mouse and human co-culture systems (Wang & Neumann, 2010; Shahraz *et al*, 2015).

In this study, we show that intravitreal application of 0.2 μg polySia avDP20 in the laser-damage mouse model reduced mononuclear phagocyte activation, vascular leakage, and membrane attack complex (MAC) deposition in humanized SIGLEC11 transgenic mice. Application of 0.2 μg polySia avDP20 in laser-treated wild-type (WT) mice similarly reduced MAC deposition. *In vitro*, polySia avDP20 inhibited the reactivity of mononuclear phagocytes via SIGLEC receptors and directly interfered with activation of the alternative complement system.

## Results

### Oligo-/polysialic acids and microglia/macrophage-specific SIGLEC11 expression are present in the retina

We first analyzed gene transcription and protein expression of the SIGLEC11 receptor in human retinas. We detected *SIGLEC11* gene transcripts in all analyzed human retinas (Fig EV1A). While no correlation between the level of *SIGLEC11* gene transcription and donor age was observed, there was a certain degree of inter-individual variation between the different human retinal samples. Furthermore, we performed immunohistochemistry on human retinal cross sections using a SIGLEC11-specific antibody (Fig EV1B). We detected SIGLEC11 mainly on ionized calcium-binding adapter molecule 1 (Iba1)-positive microglial cells (Fig EV1B). As SIGLEC11 binds to α2.8-linked oligoSia and polySia (Hayakawa *et al*, 2005; Shahraz *et al*, 2015), we next analyzed potential SIGLEC11 ligands in the human retina by immunohistochemical staining with different antibodies against polySia (CLONE 2-2B and CLONE 12F8) as well as oligoSia (CLONE 105). Intense immunoreactivity for polySia was detected in the inner plexiform layer and the ganglion cell layer as well as in the outer nuclear layer and the outer plexiform layer (Fig EV1D and E). In contrast, oligoSia was regularly distributed in a speckled manner throughout the outer and inner plexiform layer (Fig EV1F). As *SIGLEC11* is a lineage-specific gene with selective expression on human microglia (Hayakawa *et al*, 2005) without a direct homologue in the mouse, we generated and characterized a humanized transgenic mouse expressing SIGLEC11 under the macrophage/microglia-specific Iba1 promoter (Appendix Figs S1 and S2, and Appendix Table S1). *SIGLEC11* gene transcripts were present in the retinas of transgenic mice (Fig EV2A). Transcription levels were higher than compared to human retinas (Fig EV2A). Flow cytometry analysis of mouse retinas then showed that a subset of CD11b-positive and CD45-positive cells expressed SIGLEC11 protein (Fig EV2B). We next studied the retinal expression of oligoSia and polySia in SIGLEC11 transgenic mice in relation to Iba1-positive microglia (Fig EV2C–F). Immunohistochemical staining with the polySia-specific antibodies revealed a uniform immunoreactivity pattern for polySia throughout all retinal layers (Fig EV2D and E). The oligoSia-specific antibody showed a rather faint, dotted staining in all retinal layers (Fig EV2F).

These data demonstrate the presence of SIGLEC11 and the ligands oligo-/polysialic acids in human and SIGLEC11 transgenic mouse retinas.

---

**Figure 1.  Inhibition of microglia/macrophage reactivity after intravitreal injection of polySia avDP20 in mice with retinal laser lesion.**

A  Confocal images show Iba1-immunoreactive microglia and macrophages in RPE/choroid whole mount preparations 48 h after laser coagulation and intravitreal application of low (0.2 μg) and high (3 μg) dose of polySia avDP20 or PBS vehicle. Laser spots of vehicle-injected SIGLEC11 transgenic (tg) and wild-type control animals revealed strong accumulation of activated microglia and macrophages, which was effectively decreased in polySia avDP20-injected SIGLEC11 tg animals and to a lower extent in wild-type controls. Representative images out of at least three independent experiments are shown. Scale bar: 50 μm.

B  Quantification of the average pixel intensity of Iba1-positive area in laser spots reflects accumulation of reactive microglia/macrophages on RPE/choroid whole mounts. In comparison with the accumulation of reactive microglia/macrophages in control animals, there is a reduced pixel intensity found in polySia avDP20-treated SIGLEC11 tg animals. Notably, treatment with a high polySia avDP20 dose (3 μg) also reduced accumulation of microglia/macrophages in laser spots of wild-type animals. Data show mean ± SD. WT PBS ($n = 24$ spots), WT 0.2 μg Sia ($n = 37$ spots), WT 3 μg Sia ($n = 20$ spots), SIGLEC11 tg PBS ($n = 25$ spots), SIGLEC11 tg 0.2 μg Sia ($n = 35$ spots), SIGLEC11 tg 3 μg Sia ($n = 31$ spots); WT PBS vs. WT 3 μg Sia *$P = 0.0147$, WT 0.2 μg Sia vs. SIGLEC11 tg 0.2 μg Sia **$P = 0.0046$, SIGLEC11 tg PBS vs. SIGLEC11 tg 0.2 μg Sia **$P = 0.0071$, SIGLEC11 tg PBS vs. SIGLEC11 tg 3 μg Sia ***$P < 0.0001$, one-way ANOVA followed by Fisher's LSD.

C  Confocal images of retinal whole mounts show Iba1-immunoreactive microglial cells. Retinal microglia of polySia avDP20-treated SIGLEC11 tg mice had a more ramified microglial morphology in the laser spot compared to PBS vehicle-treated mice. Interestingly, high polySia avDP20 dose (3 μg) also exerted a weak therapeutic effect in wild-type animals. Representative images out of at least three independent experiments are shown. Scale bar: 20 μm.

D  Percentage of retina showing activated microglial cells within the laser spots was quantified. PolySia avDP20 reduced the percentage of laser spots with activated microglia in SIGLEC11 tg animals and at high dose (3 μg) also in wild-type controls. Data show mean ± SEM. WT PBS ($n = 6$ retinas), WT 0.2 μg Sia ($n = 8$ retinas), WT 3 μg Sia ($n = 6$ retinas), SIGLEC11 tg PBS ($n = 8$ retinas), SIGLEC11 tg 0.2 μg Sia ($n = 7$ retinas), SIGLEC11 tg 3 μg Sia ($n = 6$ retinas); WT PBS vs. WT 3 μg Sia **$P = 0.0052$, WT 0.2 μg Sia vs. SIGLEC11 tg 0.2 μg Sia ***$P = 0.0005$, SIGLEC11 tg PBS vs. SIGLEC11 tg 0.2 μg Sia *$P = 0.0159$, SIGLEC11 tg PBS vs. SIGLEC11 tg 3 μg Sia **$P = 0.001$, one-way ANOVA followed by Fisher's LSD.

---

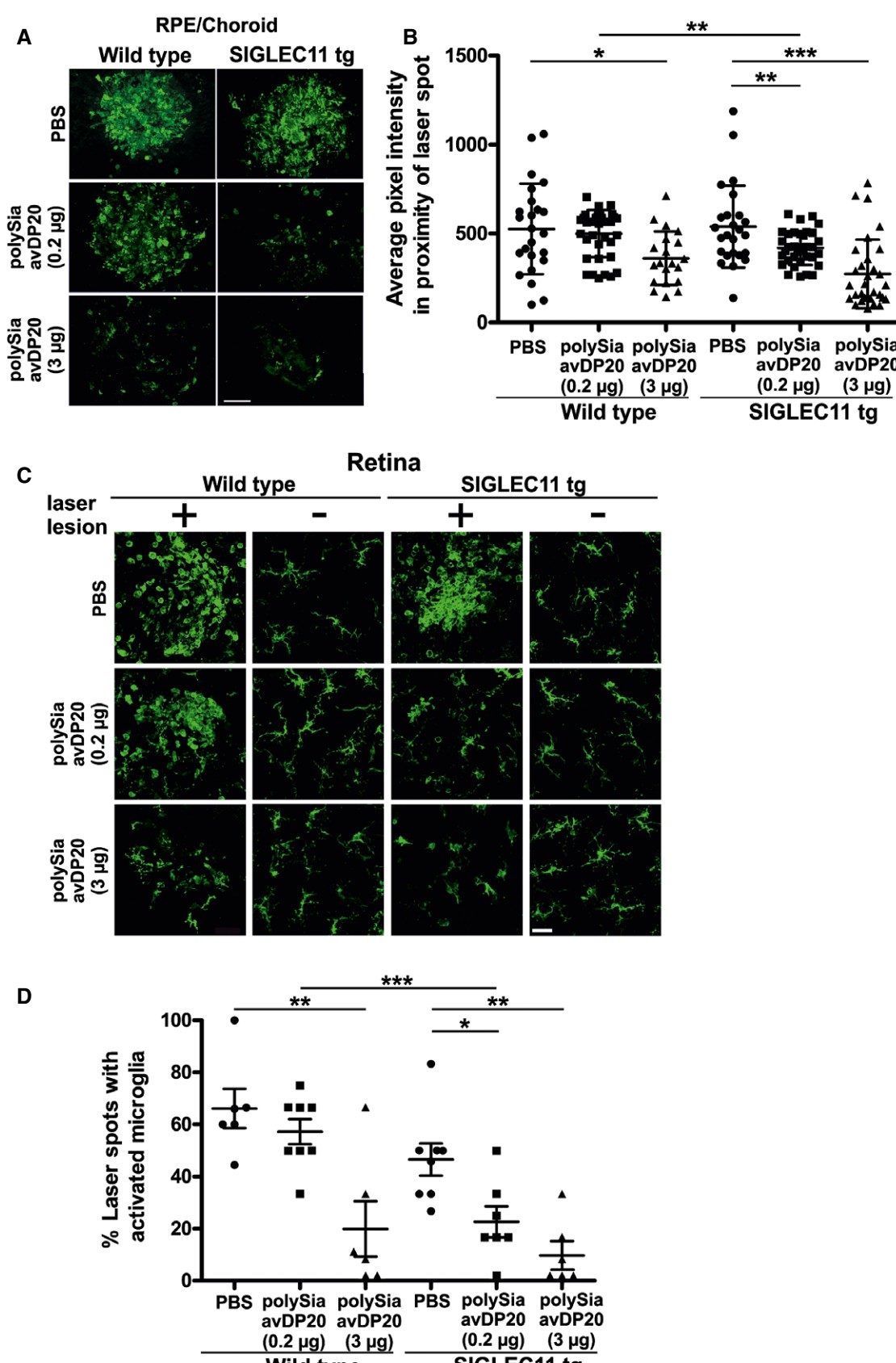

**Figure 1.**

## PolySia avDP20 prevents microglia/macrophage reactivity in the retinal laser-damage mouse model

To study the effect of polySia on immune-related features of AMD *in vivo*, we performed laser coagulation in retinas from humanized SIGLEC11 transgenic mice or littermate controls followed by intravitreal injection of polySia avDP20 at two doses (0.2 μg/eye and 3 μg/eye). Confocal images of RPE/choroid whole mounts from control mice revealed strong accumulation of reactive Iba1-positive phagocytes within the lesions at 48 h post-laser application, whereas 0.2 μg and 3 μg polySia avDP20 potently diminished subretinal microglia/macrophage accumulation in SIGLEC11 transgenic mice (Fig 1A). Quantification of the pixel intensities within laser spots reflecting accumulation of Iba1-positive phagocytes showed a relative reduction from $539.8 \pm 230.1$ in vehicle PBS-injected SIGLEC11 transgenic mice to $419.1 \pm 95.7$ ($P = 0.0071$) and $272.6 \pm 193.1$ ($P < 0.0001$) after treatment with both 0.2 and 3 μg polySia avDP20, respectively (Fig 1B). Notably, a polySia avDP20-mediated reduction in phagocyte accumulation was also observed in wild-type mice after treatment with the high dose of 3 μg polySia avDP20 ($526.2 \pm 254.8$ in PBS controls to $361.1 \pm 150.9$ in treated animals; $P = 0.0147$; Fig 1B). However, application of low-dose (0.2 μg) polySia avDP20 did not have significant effects in wild-type animals, indicating that the low-dose effects are exclusively present in SIGLEC11 transgenic mice. Confocal images of microglial cells were further obtained from retinal whole mount preparations (Fig 1C). Microglial cells were mainly reactive with an amoeboid morphology in the laser lesion of vehicle-injected and low-dose polySia avDP20-treated wild-type mice (Fig 1C). In contrast, microglia had a more ramified morphology inside the laser lesion after injection of low and high dose of polySia avDP20 into SIGLEC11 transgenic eyes (Fig 1C). After treatment with polySia avDP20, the percentage of laser spots showing activated microglia was reduced in the retinas of SIGLEC11 transgenic mice from $46.6 \pm 6.2\%$ in PBS controls to $22.6 \pm 5.9\%$ ($P = 0.0159$) and $9.7 \pm 5.4\%$ ($P = 0.001$) after injection of 0.2 and 3 μg polySia avDP20, respectively. In line with our findings in the RPE/choroid, high dose of polySia avDP20 (3 μg) also diminished the number of reactive microglia in lesioned areas of wild-type animals from $66.2 \pm 7.5\%$ in PBS controls to $19.9 \pm 10.6\%$ ($P = 0.0052$) (Fig 1D).

Taken together, low dose of polySia avDP20 reduced subretinal accumulation of microglia/macrophages and intraretinal inflammation after laser damage in SIGLEC11 transgenic mice. PolySia avDP20 also showed these effects in wild-type animals when injected at the high dose.

## PolySia avDP20 reduces vascular leakage and membrane attack complex formation after retinal laser damage

We next examined the effect of polySia avDP20 treatment on vascular damage. Vascular leakage was monitored by late-phase fundus fluorescein angiography 48 h after laser coagulation of the retina. While PBS-injected laser-damaged retinas of wild-type and SIGLEC11 transgenic mice showed prominent vascular leakage, we detected consistently reduced vascular leakage in polySia avDP20-injected SIGLEC11 transgenic animals at both doses (Fig 2A). Detailed quantitative analysis of fluorescence pixel intensity revealed vascular

leakage reduction from $117.1 \pm 24.5$ to $84.0 \pm 16.7$ ($P < 0.0001$) and $79.3 \pm 24.5$ ($P = 0.0002$) upon treatment of SIGLEC11 transgenic mice with 0.2 and 3 μg polySia avDP20, respectively (Fig 2B). Notably, wild-type mice also responded to high dose of polySia avDP20 treatment with reduced vascular leakage from $120.5 \pm 22.9$ to $98.1 \pm 21.1$ ($P = 0.0193$; Fig 2B). However, application of low-dose (0.2 μg) polySia avDP20 only attenuated vascular leakage in SIGLEC11 transgenic mice (Fig 2B).

We next investigated a potential effect of polySia avDP20 treatment on complement-mediated membrane attack complex (MAC) formation at the RPE/choroid level in the laser lesions of SIGLEC11 transgenic and wild-type mice (Fig 2C). Anti-C5b-9 immunostaining of RPE/choroidal whole mounts revealed strong MAC deposition in the laser spots of vehicle PBS-injected mice 48 h after laser damage. PolySia avDP20 treatment effectively and dose-dependently reduced C5b-9 staining intensity in the laser lesions compared to PBS controls regardless of SIGLEC11 presence (Fig 2C). Quantification of fluorescence signal intensities revealed a reduction from $48.6 \pm 7.6$ in PBS-injected to $18.0 \pm 5.3$ ($P < 0.0001$) and $11.8 \pm 6.1$ ($P < 0.0001$) in 0.2 and 3 μg polySia avDP20-treated wild-type mice, respectively, and a reduction from $49.3 \pm 18.9$ in PBS-injected to $17.1 \pm 9.6$ ($P = 0.0004$) and $12.4 \pm 4.2$ ($P < 0.0001$) in 0.2 and 3 μg polySia avDP20-treated SIGLEC11 transgenic mice, respectively (Fig 2D).

Thus, our findings in the laser-damage model demonstrated that polySia avDP20 prevented both vascular leakage and subretinal MAC formation. Inhibition of MAC occurred in a SIGLEC11-independent manner, thus suggesting a second mode of action.

## PolySia avDP20 inhibits TNFSF2 and VEGF production in reactive mononuclear phagocytes

As shown previously (Shahraz *et al*, 2015), polySia avDP20 inhibited expression of tumor necrosis factor-α (TNF-α/TNFSF2) in human THP1 macrophages upon lipopolysaccharide (LPS) treatment (Fig 3A). To confirm that this effect was mediated via SIGLEC receptors, we created SIGLEC11/16-deficient THP1 macrophages by using the CRISPR/Cas9 technology (Appendix Fig S3). LPS increased the gene transcription and protein expression of SIGLEC11 in the THP1 macrophages (Appendix Fig S4). However, polySia avDP20 failed to inhibit the LPS-induced increase in *TNFSF2* transcription and TNFSF2 protein expression in SIGLEC11/16-deficient THP1 macrophages (Fig 3A and B). Next, we analyzed the effect of polySia avDP20 on murine embryonic stem cell-derived microglia (ESdM) (Beutner *et al*, 2010). While no effect was observed at 0.15 μM polySia avDP20, a reduction in *Tnfsf2* gene transcripts was observed at 1.5 μM polySia avDP20 in LPS-stimulated ESdM (reduced from $7.18 \pm 0.98$ to $2.3 \pm 0.69$, $P = 0.001$; Fig EV3A). Since the mouse does not have a SIGLEC11 gene, we analyzed the nearest functionally related Siglec receptor, namely SiglecE. Lentiviral knockdown of SiglecE neutralized the inhibitory effect of polySia avDP20 on *Tnfsf2* gene transcription of LPS-stimulated ESdM (Fig EV3A). Since there was a clear difference in the required concentration for eliciting an anti-inflammatory response between human vs. mouse phagocytes, we performed a dose–response experiment (Fig EV3B). While polySia avDP20 showed in human THP1 macrophages expressing SIGLEC11 a half-maximal effective concentration of $EC50_{THP1} = 140$ nM on *TNFSF2* gene transcription, an approximately 10 times

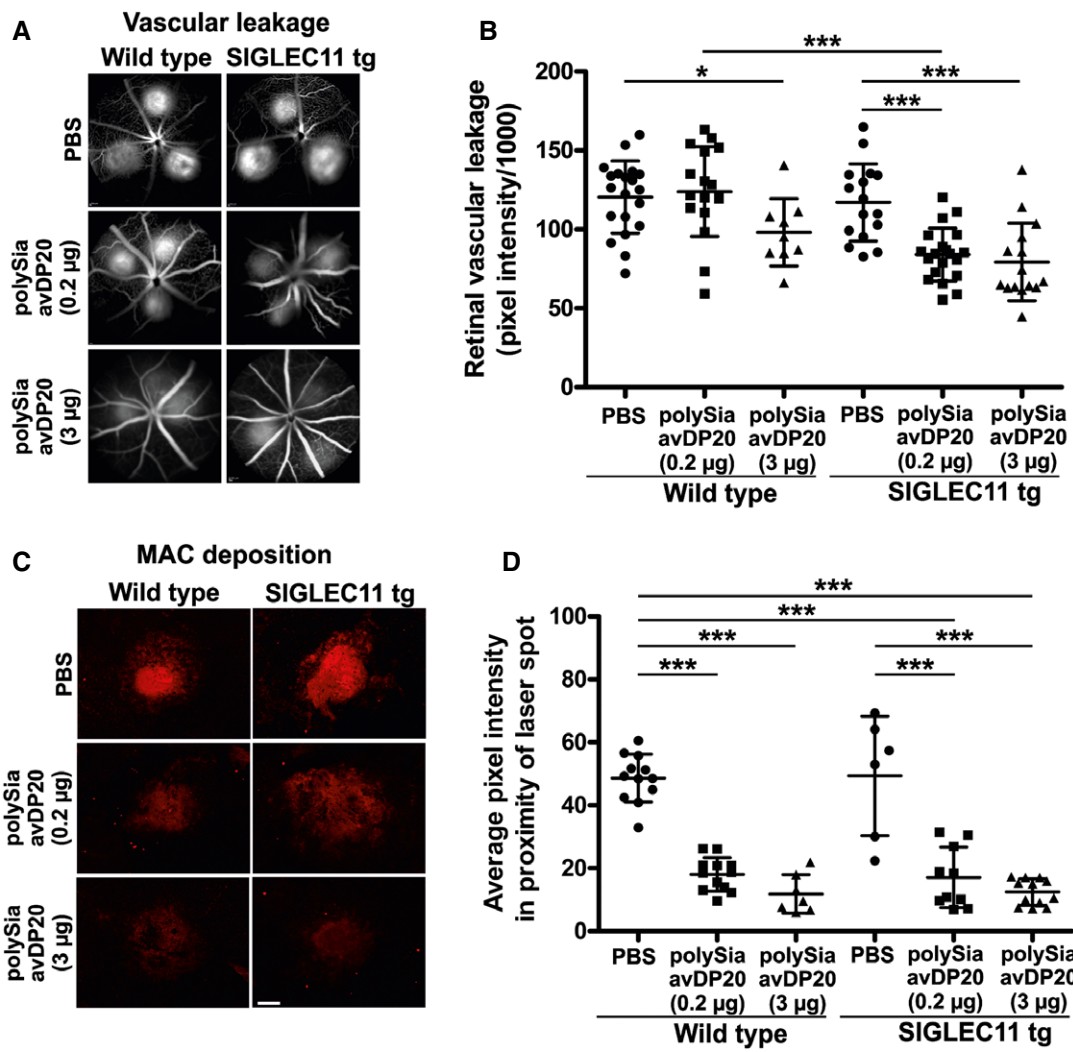

**Figure 2.  Reduced vascular leakage and membrane attack complex (MAC) formation after intravitreal application of polySia avDP20 in mice with retinal laser lesion.**

A   Fundus fluorescein angiography was performed 48 h after laser coagulation and intravitreal application of low (0.2 µg) and high (3 µg) dose of polySia avDP20 or PBS vehicle. Late-stage (10–11 min after fluorescein injection) fundus fluorescein angiography revealed that PBS-treated wild-type controls and humanized SIGLEC11 mice showed normal levels of vessel leakage, whereas polySia avDP20-treated SIGLEC11 transgenic (tg) mice had lower levels of vessel leakage compared to PBS-injected wild-type or SIGLEC11 tg mice. High polySia avDP20 dose reduced vascular leakage also in wild-type animals. Representative images out of at least eight independent experiments are shown.

B   Fundus fluorescein angiography pictures were exported from Heidelberg Eye Explorer Software, and fluorescein leakage was quantified with ImageJ software (NIH). Pixel intensities of six regions of interest per picture were quantified, and background fluorescence was subtracted. PolySia avDP20-treated SIGLEC11 tg animals showed reduced vascular leakage compared to PBS-injected wild-type control and SIGLEC11 tg mice. A reduction in vascular leakage was also observed in high-dose polySia avDP20-injected wild-type control mice. Data show mean ± SD. WT PBS ($n$ = 20 eyes), WT 0.2 µg Sia ($n$ = 17 eyes), WT 3 µg Sia ($n$ = 9 eyes), SIGLEC11 tg PBS ($n$ = 16 eyes), SIGLEC11 tg 0.2 µg Sia ($n$ = 20 eyes), SIGLEC11 tg 3 µg Sia ($n$ = 15 eyes); WT PBS vs. WT 3 µg Sia *$P$ = 0.0193, WT 0.2 µg Sia vs. SIGLEC11 tg 0.2 µg Sia ***$P$ < 0.0001, SIGLEC11 tg PBS vs. SIGLEC11 tg 0.2 µg Sia ***$P$ < 0.0001, SIGLEC11 tg PBS vs. SIGLEC11 tg 3 µg Sia ***$P$ = 0.0002, one-way ANOVA followed by Fisher's LSD.

C   Anti-C5b-9 immunostaining of RPE/choroid whole mount preparations 48 h after laser damage showed strong MAC deposition in the laser lesions of vehicle-injected controls. PolySia avDP20 treatment reduced MAC formation in a dose-dependent fashion independent of SIGLEC11 presence. Scale bar: 100 µm.

D   Quantification of C5b-9 fluorescence signal intensity in the laser lesions. In comparison with the high amount of MAC deposition in vehicle-injected controls, the pixel intensity is reduced in polySia avDP20-treated eyes in a dose-dependent fashion and regardless of SIGLEC11 presence. Data show mean ± SD. WT PBS ($n$ = 12 laser spots), WT 0.2 µg Sia ($n$ = 12 laser spots), WT 3 µg Sia ($n$ = 7 laser spots), SIGLEC11 tg PBS ($n$ = 6 laser spots), SIGLEC11 tg 0.2 µg Sia ($n$ = 10 laser spots), SIGLEC11 tg 3 µg Sia ($n$ = 12 laser spots); all statistical comparisons ***$P$ < 0.0001 except SIGLEC11 tg PBS versus SIGLEC11 tg 0.2 µg Sia ***$P$ = 0.0004, one-way ANOVA followed by Fisher's LSD.

higher concentration of polySia avDP20 (EC50$_{ESdM}$ = 1.29 µM) was required to elicit the same inhibitory activity on *Tnfsf2* transcription in mouse microglia expressing SiglecE (Fig EV3B).

Next, we analyzed the effect of polySia avDP20 on VEGF gene transcription and protein release in human THP1 macrophages. PolySia avDP20 (0.15 and 1.5 µM) inhibited the LPS-induced gene

transcription of *VEGF* (covering splice variants 121, 165, 189, and 206; Fig 3C) as well as the protein release of VEGFA (Fig 3D). In detail, *VEGF* transcription in wild-type cells was reduced from $1.7 \pm 0.16$ to $1.12 \pm 0.01$ for $0.15 \mu M$ ($P = 0.0002$) and to $1.02 \pm 0.04$ for $1.5 \mu M$ polySia avDP20 ($P < 0.0001$). Protein release of VEGFA was reduced from $573 \pm 14$ pg/ml to $475 \pm 13$ pg/ml for $0.15 \mu M$ ($P = 0.043$) and to $421 \pm 16$ pg/ml for $1.5 \mu M$ polySia avDP20 ($P = 0.0001$). This inhibitory effect of polySia avDP20 on VEGF transcription and protein release was not observed in SIGLEC11/16-deficient THP1 macrophages (Fig 3C and D).

Thus, polySia avDP20 shows SIGLEC11-dependent inhibitory effects on TNFSF2 and VEGF expression of human macrophages at relatively low concentrations, while approximately $10\times$ higher concentrations are required for anti-inflammatory effects in mouse mononuclear phagocytes expressing SiglecE.

### PolySia avDP20 prevents the phagocytosis-associated oxidative burst

Since polySia avDP20 interferes with superoxide production of human THP1 macrophages (Shahraz *et al*, 2015), we studied whether polySia avDP20 also prevents superoxide production of THP1 macrophages after incubation with debris derived from necrotic human ARPE-19 cells. PolySia avDP20 ($1.5 \mu M$) reduced the ingestion of drusen-like debris from $1 \pm 0.08$ in control cells to $0.7 \pm 0.05$ ($P = 0.028$) in polySia avDP20-treated cells (Fig 3E) and inhibited the relative superoxide release from $1.3 \pm 0.03$ after debris treatment to $0.9 \pm 0.07$ ($P = 0.001$), a level comparable to unstimulated cells, $1.0 \pm 0.03$ (Fig 3F). The inhibitory effect of polySia avDP20 on macrophage phagocytosis and radical production was absent in SIGLEC11/16-deficient human THP1 macrophages (Fig 3E and F). Interestingly, polySia avDP20 prevented superoxide release as potently as the scavengers SOD1 or Trolox (Appendix Fig S5B).

Next, we analyzed whether polySia avDP20 also inhibited phagocytosis and ROS production of mouse ESdM. Indeed, polySia avDP20 ($1.5 \mu M$) reduced the phagocytosis of microglia from $1 \pm 0.05$ to $0.6 \pm 0.03$ ($P = 0.04$; Fig EV3C and D) and completely inhibited the ROS production during the oxidative burst via SiglecE (Fig EV3E). In detail, polySia avDP20 ($1.5 \mu M$) reduced superoxide release from $1.4 \pm 0.08$ in the debris-stimulated group to $0.9 \pm 0.05$ ($P = 0.004$). Superoxide reduction was absent in the SiglecE knockdown group after debris stimulation and treatment with $1.5 \mu M$ polySia avDP20 ($1.3 \pm 0.1$, $P = 0.03$). PolySia avDP20

**Figure 3.  PolySia avDP20 inhibits TNFSF2, VEGF, and superoxide production in human macrophages and prevents activation of the alternative complement pathway.**

A  Analysis of relative *TNFSF2* gene transcription in human control and SIGLEC11/16 knockout THP1 macrophages. The levels of gene transcripts were reduced after 24 h of co-treatment with LPS (1 μg/ml) and concentrations of 0.15 and 1.5 μM of polySia avDP20 in human wild-type macrophages. No response to polySia avDP20 was detectable in the knockout line. Data show mean ± SEM. **$P < 0.01$, ANOVA followed by Bonferroni correction. Statistical analysis was done in relation to the LPS control. WT: no treatment $n = 7$ and $P < 0.0001$, polySia avDP20 1.5 μM $n = 4$ and $P = 0.0002$, LPS $n = 7$, LPS/polySia avDP20 0.15 μM $n = 3$ and $P = 0.009$, LPS/polySia avDP20 1.5 μM $n = 5$ and $P = 0.002$. SIGLEC11/16 KO: no treatment $n = 5$ and $P = 0.01$, polySia avDP20 $n = 4$ and $P = 0.122$, LPS $n = 7$, LPS/polySia avDP20 0.15 μM $n = 3$ and $P = 1.0$, LPS/polySia avDP20 1.5 μM $n = 4$ and $P = 1.0$.

B  Analysis of relative TNFSF2 protein release in human control and SIGLEC11/16 knockout THP1 macrophages. The released protein levels were reduced after 24 h of co-treatment with LPS (1 μg/ml) and concentrations of 0.15 and 1.5 μM of polySia avDP20 in human wild-type macrophages. No response to polySia avDP20 was detectable in the knockout line. Data show mean ± SEM. **$P < 0.01$, ***$P < 0.001$, ANOVA followed by Bonferroni correction. Statistical analysis was done in relation to the LPS control. WT: no treatment $n = 8$ and $P < 0.0001$, polySia avDP20 1.5 μM $n = 5$ and $P < 0.001$, LPS $n = 7$, LPS/polySia avDP20 0.15 μM $n = 5$ and $P = 0.002$, LPS/polySia avDP20 1.5 μM $n = 4$ and $P = 0.0003$. SIGLEC11/16 KO: no treatment $n = 6$ and $P < 0.001$, polySia avDP20 1.5 μM $n = 5$ and $P < 0.001$, LPS $n = 7$, LPS/polySia avDP20 0.15 μM $n = 5$ and $P = 1.0$, LPS/polySia avDP20 1.5 μM $n = 5$ and $P = 1.0$.

C  Analysis of relative *VEGFA* gene transcription in human control and SIGLEC11/16 knockout THP1 macrophages. Gene transcripts were reduced after 24 h of co-treatment with LPS (1 μg/ml) and polySia avDP20 (0.15 and 1.5 μM) in human wild-type macrophages. No response to polySia avDP20 was detectable in the knockout macrophages. Data show mean ± SEM. ***$P < 0.001$, ANOVA followed by Bonferroni correction. Statistical analysis was done in relation to the LPS control. WT: no treatment $n = 6$ and $P < 0.0001$, polySia avDP20 1.5 μM $n = 5$ and $P < 0.0001$, LPS $n = 5$, LPS/polySia avDP20 0.15 μM $n = 5$ and $P = 0.0002$, LPS/polySia avDP20 1.5 μM $n = 5$ and $P < 0.0001$. SIGLEC11/16 KO: no treatment $n = 5$ and $P = 0.022$, polySia avDP20 1.5 μM $n = 3$ and $P = 0.063$, LPS $n = 4$, LPS/polySia avDP20 0.15 μM $n = 3$ and $P = 1.0$, LPS/polySia avDP20 1.5 μM $n = 4$ and $P = 1.0$.

D  Analysis of relative VEGFA protein release in human control and SIGLEC11/16 knockout THP1 macrophages. Released protein levels were reduced after 24 h of co-treatment with LPS (1 μg/ml) and polySia avDP20 (0.15 μM and 1.5 μM) in human macrophages. Data show mean ± SEM. *$P < 0.05$, ***$P < 0.001$, ANOVA followed by Bonferroni correction. Statistical analysis was done in relation to the LPS control. WT: no treatment $n = 9$ and $P < 0.0001$, polySia avDP20 1.5 μM $n = 6$ and $P = 0.009$, LPS $n = 9$, LPS/polySia avDP20 0.15 μM $n = 6$ and $P = 0.043$, LPS/polySia avDP20 1.5 μM $n = 7$ and $P = 0.0001$. SIGLEC11/16 KO: no treatment $n = 9$ and $P = 0.349$, polySia avDP20 1.5 μM $n = 6$ and $P = 0.249$, LPS $n = 7$, LPS/polySia avDP20 0.15 μM $n = 6$ and $P = 1.0$, LPS/polySia avDP20 1.5 μM $n = 7$ and $P = 1.0$.

E  Quantification of human control and SIGLEC11/16 knockout THP1 macrophages having ingested cellular debris. PolySia avDP20 (1.5 μM) reduced the percentage of phagocytic cells having ingested drusen-like debris. No response to polySia avDP20 was detectable in the knockout macrophages. Data are presented as mean ± SEM, $n = 6$. Debris-treated WT macrophages vs. debris plus polySia avDP20-treated WT macrophages *$P = 0.028$, debris plus polySia avDP20-treated WT macrophages vs. KO macrophages ***$P = 0.00012$, ANOVA followed by Bonferroni correction.

F  Prevention of superoxide release in activated human control and SIGLEC11/16 knockout THP1 macrophages by polySia avDP20. Cultured human THP1 macrophages were stimulated with RPE cell debris or co-stimulated with debris and polySia avDP20. Addition of debris stimulated the production of superoxide. 1.5 μM polySia avDP20 completely prevented the release of superoxide induced by debris challenge. No response to polySia avDP20 was detectable in the knockout macrophages. Data are presented as mean ± SEM, $n = 6$. Untreated WT macrophages vs. debris-treated WT macrophages *$P = 0.037$, debris-treated WT macrophage vs. debris plus polySia avDP20 1.5 μM-treated WT macrophages ***$P < 0.001$, debris plus polySia avDP20-treated WT macrophages vs. KO macrophages ***$P = 0.001$, ANOVA followed by Bonferroni correction.

G  PolySia avDP20 (0.15–50 μM) was added to normal human serum (NHS) to evaluate any interference with activation of the classical complement pathway by IgM. Complement activation was determined by C3b deposition. PolySia avDP20 had no effect on C3b deposition induced by activation of the classical complement pathway. Data show mean ± SEM, $n = 3$, n.s. = not significant, ANOVA followed by Bonferroni correction.

H  PolySia avDP20 inhibits the alternative complement pathway. PolySia avDP20 (0.15–50 μM) was added to NHS and activation of the alternative pathway was induced by LPS. C3b deposition was monitored by ELISA. Data show mean ± SEM ($n = 4$; **$P < 0.01$, ***$P < 0.001$, ANOVA followed by Bonferroni correction).

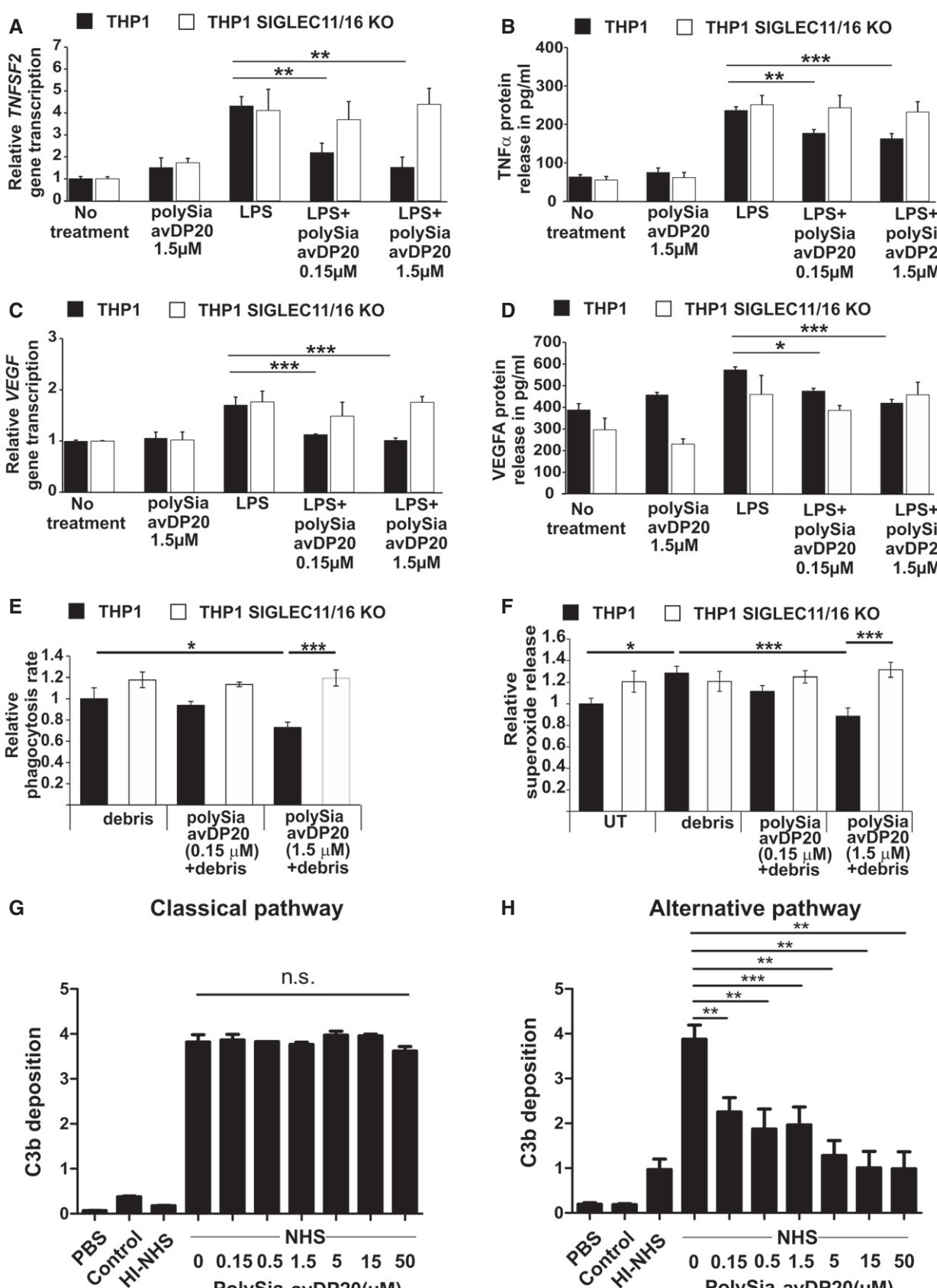

**Figure 3.**

prevented superoxide release as potently as the scavengers SOD1 and Trolox (Fig EV3F).

These data confirm that polySia avDP20 signals via SIGLEC receptors to prevent superoxide production.

### PolySia avDP20 prevents complement-mediated lysis, formation of the membrane attack complex, and activation of the alternative complement pathway

Our *in vivo* experiments revealed SIGLEC11-independent effects of polySia avDP20 on MAC deposition in the retina. This finding together with previous reports on the interaction between sialic acids and complement (Ferreira *et al*, 2010; Blaum *et al*, 2015) tempted us to investigate effects of polySia avDP20 on complement activation.

First, we used a murine cell line that is susceptible to lysis by human complement. PolySia avDP20 prevented the lysis of these cells with normal human serum (Fig EV4A). In detail, normal human serum lysed $88.9 \pm 8.4\%$ cells, while polySia avDP20 (50 μM) reduced cell lysis to $19.3 \pm 5.3\%$ ($P = 0.00019$; Fig EV4B). PolySia avDP20 protected against cell lysis induced by normal human serum even at a relatively low concentration of 2 μM (Fig EV4C). Then, we analyzed whether polySia avDP20 could also interfere with the formation of the MAC (Fig EV4D). PolySia avDP20 prevented the formation of MAC on the mouse cell line in normal human serum (Fig EV4D). In detail, cells incubated with normal human serum showed a relative C5b-9 staining intensity of $5.9 \pm 0.6$ cells, while polySia avDP20 (50 μM) reduced the MAC formation to $1.3 \pm 0.3$ ($P = 0.002$; Fig EV4E and F).

Finally, we elucidated the upstream mechanism of polySia avDP20 on alternative and classical complement activation. Complement activation was induced in normal human serum either by immunoglobulin IgM for classical pathway activation (Fig 3G) or by LPS for alternative complement activation (Fig 3H), and C3b deposition was analyzed by enzyme-linked immunosorbent assay to quantify complement activation. We found that polySia avDP20 inhibited alternative complement activation, but had no effects on classical pathway activation (Fig 3G and H). In detail, C3b deposition on LPS was reduced by 5 μM polySia avDP20 from $3.9 \pm 0.6$ to $1.3 \pm 0.7$ ($P = 0.0002$; Fig 3H). PolySia avDP20 at 0.15 μM was already sufficient to reduce alternative complement activation ($P = 0.003$), and the application of increasing concentrations revealed a dose-dependent effect.

We conclude that polySia avDP20 specifically interferes with alternative complement activation, which may present a mechanistic explanation for reduced cell lysis and MAC formation in the presence of polySia avDP20.

## Discussion

In line with our recent studies, demonstrating that interaction between sialic acid residues and SIGLEC11 contributes to decreased inflammation and neurotoxicity *in vitro* (Wang & Neumann, 2010; Shahraz *et al*, 2015), we now provide further insights into the *in vivo* effects of polysialic acid. SIGLEC11 and oligoSia and polySia were detected in the neuroretina of human *post-mortem* donors. In comparison with the human retina, the murine retina showed a more speckled and more even distribution of sialic acids as reported before for rodent retinas (Bartsch *et al*, 1990; Sawaguchi *et al*, 1999).

We carefully assessed the immunomodulatory potential of polySia avDP20 in the laser coagulation mouse model in humanized transgenic mice expressing SIGLEC11 on microglia and tissue macrophages. Laser coagulation damage induces AMD-relevant pathomechanisms including choroidal neovascularization preceded by Iba1-positive cell accumulation at the RPE/choroid (Eter *et al*, 2008; Liu *et al*, 2013; Lückoff *et al*, 2016). We demonstrate that intravitreal polySia avDP20 decreases inflammatory phagocyte accumulation in the proximity of laser lesions in SIGLEC11 transgenic mice. Reactive mononuclear phagocytes produce significant amounts of VEGF, superoxide radicals, and pro-inflammatory mediators promoting local inflammation and neovascularization (Block *et al*, 2007; Ma *et al*, 2009; Liu *et al*, 2013; Luo *et al*, 2013).

Fluorescein angiography is a standard clinical method to visualize pathological vascular leakage (Espinosa-Heidmann *et al*, 2003). We found a clear reduction in vascular leakage in polySia avDP20-treated eyes of SIGLEC11 transgenic animals 2 days after laser damage. A prior study has demonstrated that macrophage depletion by intravenous clodronate application reduces VEGF expression and laser-induced choroidal neovascularization (CNV) suggesting that phagocytes may contribute to CNV (Sakurai *et al*, 2003).

To clarify the action of polySia avDP20 on SIGLEC11-expressing phagocytes on a cellular level, we incubated THP1 macrophages with drusen-like material derived from human ARPE-19 cells. The SIGLEC11 receptor is expressed at low levels in tissue-resident macrophages and microglia (Angata *et al*, 2002; Hayakawa *et al*, 2005). Artificial cross-linking of SIGLEC11 by specific antibodies in transfected mouse macrophages recruited the phosphatase SHP-1 that transmits anti-inflammatory action (Angata *et al*, 2002). Here, we show that direct stimulation of SIGLEC11-expressing macrophages by the ligand polySia avDP20 can exert protective effects by preventing the phagocyte radical production triggered by drusen-like debris, which may to some extent explain our *in vivo* findings in the SIGLEC11 transgenic mouse.

Unexpectedly, we observed that intravitreal injection of the high dose of 3 μg polySia avDP20 also inhibits immune cell activation and vascular leakage in wild-type animals independent of SIGLEC11 function. This prompted us to test a higher concentration of polySia avDP20 also *in vitro* on cultured mouse microglia. Indeed, we found that polySia avDP20 inhibited the pro-inflammatory reactivity of mouse microglia via SiglecE receptors. A dose–response curve demonstrated that approximately a 10-fold higher concentration of polySia avDP20 was required to inhibit reactivity of mouse compared to human phagocytes. Although there is no homologue of SIGLEC11 in the mouse, SiglecE is also expressed on mouse microglia/macrophages, inhibits signaling, and interacts with α2.8-linked sialic acids (Claude *et al*, 2013; Linnartz-Gerlach *et al*, 2014). These results are further underlined by SiglecE-deficient mice, which show overt damage from radicals derived from reactive phagocytes, even leading to a reduced life span (Schwarz *et al*, 2015).

Surprisingly, we also observed reduced RPE/choroidal MAC formation in the laser lesions of polySia avDP20-treated animals at both tested concentrations independent of SIGLEC11 presence. Therefore, we also analyzed the direct effect of polySia avDP20 on the complement system, which plays a central role in the development of AMD (Ferreira *et al*, 2010; Blaum *et al*, 2015; McHarg *et al*,

2015). In a test system of human complement-susceptible murine hepatocytes, we observed diminished MAC formation and reduced cell lysis in the presence of polySia avDP20. As MAC formation is an early pathological sign of retinal laser damage, polySia avDP20 may protect against MAC deposition on RPE cells. In line with this finding, we found that polySia avDP20 inhibited alternative complement activation. It is known for decades that removal of cell surface sialic acid can trigger alternative complement activation, a mechanism possibly involving complement inhibitors (Fearon, 1978; Pangburn & Muller-Eberhard, 1978; Ferreira *et al*, 2010). Recently, Blaum *et al* (2015) reported preferential binding of CFH to trisaccharide Neu5Acα2-3Galβ1-4Glc residues, but not to α2.8-linked disialic acid. Thus, this recent study suggests that polySia avDP20 consisting of α2.8-linked polysialic acids is either no direct ligand for CFH or behaves different as compared to disialic acid.

Our results demonstrate that polySia avDP20 is a potent innate immunomodulatory biological compound and let suggest that its local application may present a plausible therapeutic strategy for retinal degenerative diseases that are linked to innate immune activation.

# Materials and Methods

Material and methods related to the mouse experiments and ethical details are given in the Appendix file.

## Human tissue

Retinal samples of donors were derived from the Eye Bank of the Department of Ophthalmology, University of Cologne, Germany. *Post-mortem* time ranged between 16 and 36 h. After removal of the cornea, the retina and the RPE were dissected and shock-frozen in liquid nitrogen or further processed for histology. The study was performed in accordance with the tenets of the Declaration of Helsinki and the Medical Research Involving Human Subjects Act (WMO) and was approved by the local ethics committee of the University Hospital in Cologne. Informed consent from all deceased individuals' family donors for tissue donation was obtained.

## RT–PCR analysis of SIGLEC11

Total RNA was extracted from total human or mouse retina according to the manufacturer's instructions using the RNeasy Mini Kit (Qiagen, Hilden, Germany). RNA integrity was assessed on the Agilent 2100 Bioanalyzer using the RNA 6000 Nano LabChip® reagent set (Agilent Technologies). RNA was quantified spectrophotometrically and stored at −80°C. First-strand cDNA synthesis was performed with the RevertAid™ H Minus First Strand cDNA synthesis Kit (Fermentas). RT–PCR was carried out to amplify intron-spanning fragments of SIGLEC11 and beta-actin or GAPDH, and PCRs were carried out using the Qiagen Taq Core kit (Qiagen) and standard PCR conditions.

## Quantitative gene transcription analysis

Total RNA was collected from either murine tissue samples or a microglia cell line as well as human THP1 macrophages via the RNeasy kit system (Qiagen). Reverse transcription of the RNA was performed using SuperScript III reverse transcriptase (Life Technologies) and random hexamer primers (Roche). Quantitative qRT–PCR with specific oligonucleotides was performed with SYBR Green PCR Master Mix (Qiagen) using the ABI 5700 Sequence Detection System (Thermo Scientific). All qRT–PCRs were running for 40 cycles with a Tm of 60°C. The $\Delta\Delta C_{\mathrm{T}}$ method with *GAPDH/gapdh* as internal standard was performed for qRT–PCR quantification.

## TNFSF2 and VEGFA protein detection by ELISA

For the detection of TNFSF2 and VEGFA release by human THP1 cells or murine microglia, Quantikine ELISA kits (R&D Systems) were used. The cells were stimulated with 1 μg/ml LPS and 0.15 or 1.5 μM polySia avDP20 for 24 h. Supernatant was harvested and processed according to the manufacturer's instructions. Optical density of the ELISA samples was determined by a spectrophotometer at a wavelength of 450 nm with a reference wavelength of 560 nm (PerkinElmer, Envision Multiplate Reader).

## Immunohistochemistry of retina and RPE/choroid whole mounts

To determine laser coagulation-dependent recruitment and activation of microglial cells, retinal whole mounts were analyzed. For retina and RPE/choroidal whole mount preparation, enucleated eye bulbs were fixed in 4% paraformaldehyde (PFA; Sigma) for 4 h at 4°C. To derive explant tissues, the fixed eye bulbs were dissected followed by overnight incubation on a rocker in a solution containing 5% Triton X-100 and 5% Tween-20 in 1× PBS to improve antigen accessibility. To reduce non-specific background staining, whole mount tissue was then incubated in BLOTTO (1% dried milk powder, 0.01% Triton X-100 in 1× PBS) for 1 h at room temperature followed by overnight incubation with antibodies directed against Iba1 (1:500, rabbit polyclonal; Wako) or C5b-9 (1:500, rabbit polyclonal; Abcam) at 4°C. After washing in 1× PBS for three times, retinal tissue was labeled with secondary antibody directed against rabbit IgG conjugated to Alexa488 or Alexa594 (1:1,000 in 1× PBS; Invitrogen) for 1 h at room temperature. Whole mount tissues were mounted on a microscope slide, embedded with fluorescent mounting medium (Dako Cytomation GmbH, Hamburg, Germany) and imaged on a Zeiss Imager M.2 equipped with an ApoTome.2. Microglial cells on retinal whole mounts were counted by two blinded analysts. Microglial cells and C5b-9 staining in RPE/choroidal laser spots were analyzed by measuring the pixel intensity of immunofluorescence within a region of interest (ROI) of 200 μm in diameter around the laser spot. All pictures have the same exposure time. Quantifications were performed by two blinded analysts.

## Immunohistochemistry of human retinal cross sections

Human retinas were dissected from donor eye bulbs and fixed with 4% formaldehyde for 4 h at room temperature. After overnight incubation in 30% sucrose in 1× PBS, retinas were embedded in optimal cutting temperature (OCT) compound (Hartenstein, Würzburg, Germany) and sliced into 12-μm-thick sections. For staining, slides were dried at room temperature for 10–15 min and washed

two times in PBS. Slices were blocked with 10% bovine serum albumin (BSA; Sigma), 5% goat serum (Invitrogen), and 0.1% Triton X-100 for 20–30 min. Slices were incubated in one of the following primary antibodies overnight at 4°C: rabbit anti-Iba1 (1:1,000; Wako), mouse anti-PSA-NCAM (1:500, polysialic acid; Millipore), rat anti-PSA (CLONE 12F8, 1:200; BD Pharmingen), or mouse anti-oligosialic acid (CLONE 105, 1:200; Invitrogen), and anti-SIGLEC11 (1:500, clone 3EH, binding up to a dilution of 1:128K to the SIGLEC11-specific peptide ISISHDNTSALE) (Shahraz *et al*, 2015). Slices were washed with PBS three times and then incubated with the corresponding Cy3-conjugated secondary antibody (Jackson) for 4 h at room temperature. After three washing steps in PBS, slices were incubated in DAPI for 15 min at room temperature and mounted with Moviol.

### Human macrophage cell culture

The human monocyte cell line THP1 derived from an acute monocytic leukemia patient was used to obtain macrophages (ATTC, USA). THP1 monocytes were cultured in medium containing RPMI (Gibco) supplemented with 10% defined fetal bovine serum (Gibco), 1% penicillin/streptomycin (Gibco), 1% L-glutamine (Gibco), and 1% sodium pyruvate (Gibco). At least 1 week before the experiment, THP1 monocytes were moved to THP1 differentiation medium containing RPMI with 1% penicillin/Streptomycin, 1% L-glutamine, 1% sodium pyruvate plus 1% N2 supplement (Gibco), and 1% chicken serum (Gibco). To differentiate THP1 cells into a macrophage phenotype, the cells were treated in differentiation medium with 10 ng/ml phorbol-12-myristate-13-acetate (PMA; Sigma) for 48 h. Afterward, cells were washed two times with 37°C warm medium and were kept in PMA-free differentiation medium for 48 h. For the experiments, serum-free medium (differentiation medium without chicken serum) was applied to the macrophages.

### Mouse microglia cell line culture

Embryonic stem cell-derived microglia (ESdM) were previously shown to behave similar to primary cultured microglia (Beutner *et al*, 2013). ESdM were cultured in N2 medium composed of DMEM–F12 (Gibco, Invitrogen), supplemented with N2 supplement, 0.48 mM L-glutamine, and 100 g/ml penicillin/streptomycin solution at 37°C and 5% $CO_2$.

### Phagocytosis assays

To obtain drusen-like retinal material, human retinal pigment epithelial (ARPE-19) cells were treated with 80 nM okadaic acid (Sigma) for 24 h at 37°C and 5% $CO_2$, centrifuged, and washed three times with PBS, and the pellet was frozen at −20°C. After thawing, cellular debris was incubated with 1 μM "DiI Derivatives for Long-Term cellular Labeling" for 5 min at 37°C followed by an incubation time of 15 min at 4°C and three washing steps. Human macrophages or mouse microglial cells were incubated with 5 μg/μl debris for 1.5 h at 37°C in the presence or absence of 0.15 μM or 1.5 μM polySia avDP20 and subsequently washed three times with PBS. Cells were fixed with 4% PFA, washed three times, and blocked (10% BSA, 5% normal goat serum, 0.1% Triton X-100) for

1 h. Cells were then incubated with a primary antibody directed against CD11b (BD Pharmingen) or Iba1 (Wako) followed by a secondary Alexa488-conjugated antibody directed against rat IgG (Invitrogen) or Alexa488-conjugated antibody directed against rabbit IgG (Invitrogen) for 2 h at room temperature. Cells were mounted in Moviol. For analysis, images were randomly obtained with a confocal laser scanning microscope and the fluorescently labeled debris was visualized inside phagocytic cells by 3D-reconstruction (Fluoview 1000; Olympus). Quantification was performed using ImageJ software comparing phagocytosing to non-phagocytosing cells.

### Detection of ROS during phagocytosis of drusen-like debris

To measure the relative production of superoxide by phagocytes, human THP1 monocyte cells and mouse microglial cells were seeded in four-chamber culture dishes as described previously (Shahraz *et al*, 2015). After 48 h, cells were treated with 5 μg/μl debris for 15 min with or without 1-h polySia avDP20 pre-incubation. To test the antioxidant effect of SOD1 or Trolox as positive controls, phagocyte cells were pre-incubated for 1 h with either 60 U/ml SOD1 (Serva) or 40 μM Trolox (Cayman), and then, debris was added. Afterward, cells were washed two times with Krebs–HEPES buffer and incubated for 15 min with 30 μM DHE solution (diluted in Krebs–HEPES buffer). Finally, cells were washed two times with Krebs–HEPES buffer and fixed for 15 min with 0.25% glutaraldehyde and 4% PFA. In total, three images were randomly collected per experimental group by confocal laser scanning microscopy (Fluoview 1000; Olympus). All cells of the collected images were analyzed by ImageJ software (NIH).

### Laser coagulation and intravitreal injections

To induce laser coagulation damage, a slit-lamp-mounted diode laser system (Viridis; Quantel Medical, France) was used to deliver three peripheral laser burns to the retinas (laser settings: 125 mW power, 100 ms duration, 100 μm spot diameter). These settings generate subretinal damage eliciting profound microgliosis and migration of phagocytes to the laser spot. Eyes that developed vitreous bleeding or showed significant cataract or keratopathy formation were excluded from the analysis to avoid inefficient laser coagulation. Directly afterward, the mice were injected intravitreally with polySia avDP20 (0.2 or 3 μg/eye) or a PBS vehicle control by incising the sclera at the pars plana with a 32G needle followed by inserting the blunt end injection needle (Hamilton, Switzerland). After the injection procedure, eyes were rinsed with antibiotic eye drops to avoid any ocular inflammation and covered with 2% Methocel (OmniVision) to avoid over-drying of the cornea. Retinal and RPE/choroid tissue was analyzed 48 h after the laser lesion by immunohistochemistry.

### Spectral domain optical coherence tomography (SD-OCT) and fundus fluorescein angiography

Successful laser coagulation was determined by SD-OCT 48 h after laser coagulation on a Spectralis HRA + OCT device (Heidelberg Engineering GmbH, Dossenheim, Germany). Mice were placed on a custom mounting platform for SD-OCT measurements

($\lambda$ = 870 nm; acquisition speed, 40,000 A-scans per second; average images per scan, 24). SD-OCT volume scans of 61 B-scans with 70 $\mu$m distance between B-scans (human dimension) were performed on the laser spots; this corresponds to 23.33 $\mu$m distance between B-scans for murine eyes. Artificial tears were used to avoid dehydration of the cornea. To determine the inflammation-mediated vascular leakage, fluorescein angiography was performed directly after SD-OCT imaging of the retina. Late-stage angiography pictures were taken 10–11 min after fluorescein injection on a Spectralis HRA + OCT device (Heidelberg Engineering GmbH) to visualize vessel leakage. Angiography images were exported from Heidelberg Eye Explorer Software as jpeg files. To quantitatively determine vessel leakage, pixel intensities around laser spots in pictures obtained by fundus fluorescein angiography were analyzed. Using ImageJ software, pixel intensities of six defined regions of interest (ROIs) were measured per retina (two ROIs per laser spot), omitting areas with subjacent retinal vessels. Background fluorescence was subtracted.

### Analysis of the effects of polySia avDP20 on cell lysis by human serum complement

Complement-mediated lysis was analyzed by using murine hepatoma cells (Hepa-1c1c7; Sigma) that are susceptible to human complement. Hepa-1c1c7 cells were cultivated in alpha-MEM with 10% fetal calf serum (FCS), 1% L-glutamine, and 1% penicillin/streptomycin. To study the complement-mediated cellular lysis, cells were seeded at high density (80% confluence), serum-starved (2% FCS) for three further days, and then washed with PBS and treated with 0.25% trypsin. Subsequently, cells were incubated in lysis buffer (gelatin-veronal buffer with calcium and magnesium; $GVB^{++}$, Complement Technology). For lysis, 0.8% normal human serum (NHS; Sigma) or 0.8% heat-inactivated NHS (complement-inactivated by heating for 1 h at 56°C) was pre-incubated for 1 h at 37°C and then added to the cultured cells for 1 h at 37°C under continuous rotation. To test the inhibition of complement-mediated lysis by polySia avDP20, different concentrations of polySia avDP20 were added to the NHS prior to pre-incubation. Successful lysis, that is, cell rupture, was quantified by flow cytometry. Cells were incubated with propidium iodide (PI), processed in the FACSCalibur device (BD), and analyzed with FlowJo software (BD).

### Analysis of effects of polySia avDP20 on membrane attack complex (MAC) formation

Membrane attack complex (MAC) formation under influence of polySia avDP20 was also studied in the Hepa-1c1c7 cells (see above) after incubation with human serum. Hepa-1c1c7 cells were cultured in alpha-MEM with 2% FCS for 3 days. Cells were then washed with PBS and incubated with pre-heated (1 h at 37°C) 10% NHS (Sigma) for 5 min in lysis buffer ($GVB^{++}$; Complement Technology). Control cells were treated with heat-inactivated NHS. PolySia avDP20 effect on MAC formation was tested by co-stimulation of the above-mentioned cell culture setups with different concentrations of polySia avDP20. The polySia avDP20 was added to the NHS prior to the pre-incubation. This step was followed by a PBS washing step and fixation with 4% PFA. MAC formation was detected

**The paper explained**

**Problem**

Age-related macular degeneration (AMD) is a leading cause of legal blindness and associated with chronic activation of retinal immune cells and the complement system. Sialic acids inhibit human phagocyte reactivity and complement activation via the inhibitory sialic acid-binding immunoglobulin-like lectin-11 (SIGLEC11) receptor. Here, we hypothesized that purified polysialic acid may reduce phagocyte reactivity and mediate retinal protection in a humanized mouse model of AMD-like laser-induced retinal injury.

**Results**

Intravitreal injection of low molecular weight polysialic acid with an average degree of polymerization 20 (polySia avDP20) in humanized transgenic mice expressing SIGLEC11 on phagocytes prevented innate immune activation, vascular leakage, and complement-mediated membrane attack complex formation in the laser-induced AMD-like retinal damage model. PolySia avDP20 acted synergistically on the innate immune system via SIGLEC-mediated inhibition of phagocyte production of radicals and interference with activation of the alternative complement pathway.

**Impact**

We could demonstrate that polySia avDP20 reduces pathological features of AMD-like retinal degeneration by inhibiting innate immune cell reactivity and complement activation. These findings suggest that polySia avDP20 is a novel promising candidate for AMD therapy by inhibiting the damaging effects of innate immune activation.

and quantified by staining with a monoclonal mouse anti-human C5b-9 antibody (1:100; Abcam) as the primary antibody for 2.5 h at 37°C and a Cy3-conjugated goat anti-mouse antibody as the secondary antibody (1:200; Dianova) for 1.5 h at room temperature. Cells were incubated with DAPI (1:10,000) and mounted with Moviol. Cells were visualized by confocal microscopy (Olympus) and quantified with the ImageJ software (NIH).

### Complement pathway activation

Complement activation assays were performed as previously described (Eberhardt *et al*, 2013). NHS was diluted in Mg–EGTA buffer (20 mM HEPES, 144 mM NaCl, 7 mM $MgCl_2$, and 10 mM EGTA, pH 7.4) for specific induction of the alternative pathway or $GVB^{++}$ buffer (Complement Technologies) for induction of the classical pathway. NHS (20% for alternative pathway activation and 1% for the classical pathway) was pre-incubated for 15 min at 37°C with PSA (0.15–50 $\mu$M) and added to microtiter wells pre-coated with either LPS (10 $\mu$g/ml) for the induction of the alternative pathway or IgM (2 $\mu$g/ml) for the classical pathway and incubated for 1 h at 37°C. Complement activation was measured by ELISA using anti-human C3b (Fitzgerald).

### Statistical analysis

Descriptive statistics, including means, standard errors of the mean (SEM), or standard deviations (SD), were computed at each time point for each experimental and control group as indicated in the figure legends. Exact *P*-values and sample sizes are listed in Appendix Table S2. Data were analyzed by unpaired *t*-test for

experiments with two groups only and by one-way ANOVA for more than two groups followed by Bonferroni correction or by Fisher's LSD test using SPSS 23 software as indicated in the figure legends.

**Expanded View** for this article is available online.

## Acknowledgements

This project was supported by the Deutsche Forschungsgemeinschaft (DFG-KFO177, DFG-SFB704, DFG-NE507/14-1, DFG-FOR2240, DFG-LA1203/9-1, DFG-Sk46/2-2), the Hertie Foundation, and the Hans und Marlies Stock-Foundation. HN and BL-G are members of the DFG-funded excellence cluster ImmunoSensation (EXC 1023). MK received support by the Koeln Fortune Program/Faculty of Medicine, University of Cologne. This project was kindly supported by a grant from Bayer Pharma AG ("from targets to novel drugs" 2014-08-1139) and by governmental grants (NRW-Patent-Validierung; BMBF VIP+ 03VP00271). We thank Jessica Reinartz, Rita Jietou, and Eva Scheiffert for excellent technical support of cultures and molecular biology. Furthermore, we thank Prof. Thomas Scheper for polysialic acids and Prof. Volkmar Gieselmann for his support in the isolation of polysialic acids. Iba1 promoter (Hirasawa *et al*, 2005) was kindly provided by Dr. S. Kohsaka (kohsaka@ncnp.go.jp), Department of Neurochemistry, National Institute of Neuroscience, Tokyo, Japan. We thank Dr. A. Varki for helpful discussions.

## Author contributions

MK, JK, CS, TL, and HN designed research; MK, JK, AA, AC, AS, BL-G, YL, AL, KD, JC, YW, JA, TS, VH, and SF performed research; MK, JK, AA, AC, AS, BL-G, YL, AL, KD, JC, YW, JA, and SF analyzed the data; MK, TL, and HN wrote the manuscript.

## Conflict of interest

Dr. Marcus Karlstetter, Dr. Anahita Shahraz, Dr. Jens Kopatz, Dr. Harald Neumann, and Dr. Thomas Langmann are named inventors on patent applications related to the use of polysialic acid for neurodegenerative diseases filed by the universities of Bonn and Cologne. Dr. Marcus Karlstetter is an employee of Bayer Pharma AG since September 1, 2015. Dr. Sascha Fauser is an employee of Roche Pharma AG since April 1, 2016.

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
