## [Review Process File · EMBO Molecular Medicine]

Polysialic acid blocks mononuclear phagocyte reactivity, inhibits complement activation and protects from vascular damage in the retina

Marcus Karlstetter, Jens Kopatz, Alexander Aslanidis, Anahita Shahraz, Albert Caramoy, Bettina Linnartz-Gerlach, Yuchen Lin, Anika Lückhoff, Sascha Fauser, Katharina Düker, Janine Claude, Yiner Wang, Johannes Ackermann, Tobias Schmidt, Veit Hornung, Christine Skerka, Thomas Langmann, and Harald Neumann

Corresponding authors: Thomas Langmann, University of Cologne
Harald Neumann, University of Bonn

Review timeline:

Submission date:	24 May 2016
Editorial Decision:	22 June 2016
Revision received:	06 September 2016
Editorial Decision:	31 October 2016
Revision received:	15 November 2016
Editorial Decision:	21 November 2016
Revision received:	21 November 2016
Accepted:	22 November 2016

Transaction Report:

Editor: Céline Carret

1st Editorial Decision

22 June 2016

Thank you for the submission of your manuscript to EMBO Molecular Medicine. We have now heard back from the two referees whom we asked to evaluate your manuscript. Although the referees find the study to be of potential interest, they also raise a number of concerns that need to be addressed in the next final version of your article.

You will see from the comments below that while the referees find the study of interest, they are not fully convinced that the conclusions are well supported at this stage. They particularly would like to see a more detailed analysis of the mode of action especially as the *in vitro* data do not seem to match well the *in vivo* effects. Besides, while Siglec11 is the main topic of the study, further mice analysis focus on Siglec9-for which the relevance is not clear. Overall, we would like to give you a chance to address all issues raised in a revised article within the time constraints outlined below. Please note that it is EMBO Molecular Medicine policy to allow only a single round of revision and that, as acceptance or rejection of the manuscript will depend on another round of review, your responses should be as complete as possible.

Revised manuscripts should be submitted within three months of a request for revision; they will

otherwise be treated as new submissions, except under exceptional circumstances in which a short extension is obtained from the editor.

I look forward to seeing a revised form of your manuscript as soon as possible.

***** Reviewer's comments *****

Referee #1 (Comments on Novelty/Model System):

Mostly acceptable although I do have some concerns about the transgenic mouse (see review comments)

Referee #1 (Remarks):

General comments

This is a well performed comprehensive study exploring the effects of low molecular weight polysialic acid on a wound healing response in the mouse retina after laser injury. The authors suggest that treatment with this agent may have a beneficial effect on innate inflammatory responses in the eye. They demonstrate a significant suppressive effect but also indicate that the mechanism is not directly linked to inhibition of Sigle11 or Siglec E positive macrophages but may be more related to inhibition of reactive oxygen species production and complement activation. In this respect anti-inflammatory effects of polysialic acid are non-specific and indeed there may be other un-explored mechanisms for instance, as prevention of acute phase response, fibrin induction, kininogen activation, plus many others not explored here. The authors therefore appear to have confirmed a general anti-inflammatory effect of cell-associated sialic acid.

Specific comments

What effect does introduction of a transgene under the control of Iba-1 have on the physiological function of mouse myeloid cells?

There are some typographical errors.

Leakage persists in the mice fundi at a lower level compared to controls, in the treated mice.

In humans, is Siglec 11 modified by the activation status of the macrophages?

Referee #2 (Comments on Novelty/Model System):

Lack of an in vivo effect suggests that either the in vitro effect is irrelevant to the physiological context or that the in vivo model does not resemble cellular events accurately. The laser-induced model is useful for assessing inflammatory response to thermal energy but does not reflect events occurring in macular degeneration.

Referee #2 (Remarks):

In this study Karlstetter et al explore the role of Siglec11 in modulating mononuclear cell activation in vitro and in vivo, using laser coagulation as a model of AMD. The authors nicely show that Siglec11, along with some of its ligands, are expressed in human retinas. I particularly liked their in vitro experiments using the THP-1 monocyte cell-line; the authors show strong evidence that Siglec11-ligand interactions suppress secretion of pro-inflammatory molecules and reactive oxygen species, and suppress phagocytic function in these cells. It is also shown that intravitreal injection of poly-sialic acid (polySia avDP20) robustly reduced mononuclear cell infiltration and activation, as well as vascular abnormalities following laser photocoagulation, which could potentially be of therapeutic benefit for diseases such as AMD, where chronic inflammation may occur. The authors claim these in vivo effects are mediated by interactions of polySia avDP20 with siglec11. They use a novel humanized mouse, expressing human siglec11, to elucidate the function of this molecule in vivo. The study is nicely designed and carefully executed, but a major weakness is that the in vivo data simply does not support a role for Siglec11 in any of the outcomes measured. Intravitreal injection of polySia avDP20 has a comparable effect on all outcomes measured in WT and Siglec11 mice.

Major comments

1. The authors claim that polySia avDP20 has an anti-inflammatory effect and reduces vascular leakage in a Siglec11 dependent manner. The data they show do not support this:

- Fig 1 B: polySia avDP20 leads to an identical reduction in Iba1 stain in both WT and Siglec11 tg mice. The effect of polySia avDP20 is in no way different in WT vs Siglec11 tg mice.

- Fig 1D: There is a higher number of microglia per laser spot in the retina of WT mice when compared to the Siglec11 Tg mice after injection of the lower dose of polySia avDP20 (0.2ug). However, Siglec11 Tg mice have much lower levels of microglia per laser spot to begin with (when compared to WT after PBS injection). There does not seem to be a real difference in the effect of 0.2ug of polySia avDP20 vs PBS between the two genotypes. Additionally, when looking at the higher dose of polySia avDP20 (3ug), the effect of intravitreal injection on the number of microglia seems identical, with the exception of one (very large) outlier in the WT group.

- Fig 2B: similar to the previous data, the high dose of polySia avDP20 does not seem all that different between WT and Siglec11 Tg mice. In contrast to data in figure 1, here the lower dose of 0.2ug seems to have a bigger effect in the Siglec11 Tg mice vs WT.

If Siglec11 were to mediate the effects of polySia avDP20 one would expect much larger differences between WT and transgenic mice. If anything, data from figure 2B suggests that Siglec11 may contribute to the effects of polySia avDP20 on vascular leakage, but in no way do these data support Siglec11 as the main mediator of the in vivo effects.

2. Figure 3. In this paper the authors explore the function of Siglec11, for which there is no mouse homologue. However, in the experiments presented in EV Fig 3, instead of using microglia from their transgenic mouse expressing Siglec11, the authors knock-out SiglecE (or Siglec 9). While Siglec9 and 11 belong to the same family they are different molecules. In addition, humans express Siglec9. It is not clear to me why the authors studied SiglecE/Siglec9 in these in vitro experiments, but not in vivo. Are the effects they are seeing in both WT and Siglec11 Tg mice mediated by Siglec9? Why have the authors not looked at Siglec9 in their in vivo studies? The role of this Siglec in their study should at the very least be discussed.

3. EV Fig 1, panel F. The two halves of panel F very clearly do not belong to the same section/stain.

4. EV Fig 2B. Here the authors perform flow cytometric analysis of brain tissue (which is never mentioned in the main text). It is not clear to me why the authors are examining expression of Siglec11 in the brain and why this is included with retina data.

Minor comments

1. Page 3. Reference 'FritscheIgl et al;' needs to be corrected.

2. Page 5 and EV Fig.2A. The authors claim comparable levels of Siglec11 gene transcripts are detected in the retina of humans and transgenic mice. The gel shown in EV figure 2 appears to show lower levels of expression in the human retinas compared to mouse. The authors should add a graph with the normalized quantification of Siglec11 transcript levels.

3. Page 9. "as well as the protein release of VEGFA (Fig. 6D)." Do the authors mean Figure 3B?

4. Figure 3 is hard to read, please add labels (e.g. B, D) to the two top right panels (Tnfa protein and VEGFA protein)

5. The authors should consider changing the graphs in Figure 3 to compare treatments and not genotype (as is). In the text the authors mostly describe differences between treatment in WT THPs, which are no longer observed in the THP1 Siglec11/16 KO, rather than directly compare e.g. the effect of LPS on WT vs KO THP cells. It would make it easier for the reader to follow the text if the graphs were changed accordingly.

1st Revision - authors' response

06 September 2016

Referee #1 (Comments on Novelty/Model System):

Mostly acceptable although I do have some concerns about the transgenic mouse (see review comments)

Referee #1 (Remarks):

General comments

This is a well-performed comprehensive study exploring the effects of low molecular weight polysialic acid on a wound healing response in the mouse retina after laser injury. The authors suggest that treatment with this agent may have a beneficial effect on innate inflammatory responses in the eye. They demonstrate a significant suppressive effect but also indicate that the mechanism is not directly linked to inhibition of Siglec-11 or Siglec-E positive macrophages but may be more related to inhibition of reactive oxygen species production and complement activation. In this respect anti-inflammatory effects of polysialic acid are non-specific and indeed there may be other un-explored mechanisms for instance, as prevention of acute phase response, fibrin induction, kininogen activation, plus many others not explored here. The authors therefore appear to have confirmed a general anti-inflammatory effect of cell-associated sialic acid.

*We don't believe that the anti-inflammatory effects of polySia avDP20 are non-specific. So far, we have not determined the binding affinity of polySia avDP20 to SIGLEC11, but we would like to point out that the effective concentration we are using is relative low and in a nanomolar range. Recently, we showed that polySia avDP20 has a half maximal anti-inflammatory efficacy of EC50 = 140 nM via SIGLEC11 in human macrophages, while mono-, tri- and hexa-sialic acid were lacking biological activity, suggesting that this effect is mediated by a relatively specific interaction of polySia avDP20 with SIGLEC11 (Shahraz et al. 2015, Sci. Rep. Nov 19;5:16800). Furthermore, we show here that polySia avDP20 also interferes with the complement system by inhibiting complement-mediated lysis at a half maximal effective concentration of EC50 = 700 nM. Again, this effect is not observed with sialic acid monomers, suggesting that this is another relative specific interaction.

The observation that sialic acid containing glycans have specific protein interactors is also in agreement with recent publications demonstrating that sialic acid requires certain glycosidically linked carbohydrates to specifically interact with proteins. Complement factor H preferentially binds to tri-saccharide Neu5Aca2-3Galβ1-4Glc residues, but not to α2.8-linked di-sialic acids (Blaum et al. 2015, Chem. Biol. Jan;11(1):77-82). SIGLEC8 simultaneously recognizes a terminal N-acetylneuraminic acid (sialic acid) and an underlying 6-O-sulfated galactose, yielding a tight and unique specificity (Pröpster et al. 2016. Proc Natl Acad Sci U S A. Jun 29. pii: 201602214). Thus, sialic acid as monomer does not show any general anti-inflammatory effect, but specifically acts in conjunction with other carbohydrate residues on innate immune-related proteins.*

Specific comments

What effect does introduction of a transgene under the control of Iba-1 have on the physiological function of mouse myeloid cells?

*The Iba-1 promoter used to create our transgenic mice has been used before to drive expression of the reporter EGFP (Hirasawa T, et al. 2005, J Neurosci Res 81: 357-62). These mice have been used in several studies without obvious phenotype. In our transgenic mice the Iba-1 promoter controls expression of human SIGLEC11. The mice showed normal breeding, typical weight and behavior. We observed *SIGLEC11* gene transcription in the retina, brain, and spinal cord as well as in spleen and liver (Supplementary Figure 1). We did not find any *SIGLEC11* transcription or protein expression in peripheral blood cells (Supplementary Figure 1).

To further exclude any other side effects on Iba-1 positive cells, we stained Iba-1 and CD68 in the brain and spleen. Cell numbers of Iba-1 and CD68 positive cells were unchanged in the SIGLEC11 transgenic mice compared to littermate control mice (Supplementary Figure 2). In concordance, the characteristic ramified shape of microglial cells in wild type retina and brain tissue was observed throughout all analyzed samples leading us to the conclusion that recombinant SIGLEC11 expression does not alter the immune status of microglial cells.

Furthermore, we determined the gene transcription of tumor necrosis factor-α, interleukin-1b, and Iba-1 in the brain and spleen of SIGLEC11 transgenic compared to littermate control mice with and without challenge with LPS. We did not find any changes in the gene transcription of tumor necrosis factor-α, interleukin-1b, and Iba-1 in the SIGLEC11 transgenic mice (Supplementary Figure 2).*

There are some typographical errors.

Now corrected.

Leakage persists in the mice fundi at a lower level compared to controls, in the treated mice.

We agree with the reviewer that mice treated intravitreally with 3 mg polySia avDP20 still show some residual leakage. Thus, polySia avDP20 only reduced the leakage, but did not fully abolish it. We more carefully described this finding in the text (see page 7).

In humans, is Siglec11 modified by the activation status of the macrophages?

We performed additional experiments with THP1 macrophages by analyzing SIGLEC11 gene transcription by qRT-PCR and protein expression by FACS analysis after treatment with different inflammatory stimuli. We observed a significant upregulation of SIGLEC11 gene transcription as well as protein expression after stimulation with lipopolysaccharides. However, downregulation of SIGLEC11 receptor transcripts was not observed in any condition. Therefore, SIGLEC11 is likely to be higher expressed on inflammatory macrophages (Supplementary Figure 4).

Referee #2 (Comments on Novelty/Model System):

Lack of an *in vivo* effect suggests that either the *in vitro* effect is irrelevant to the physiological context or that the *in vivo* model does not resemble cellular events accurately. The laser-induced model is useful for assessing inflammatory response to thermal energy but does not reflect events occurring in macular degeneration.

We agree with the reviewer that each animal model has limitations in respect to translation into clinics. Nevertheless, successful testing of novel treatment strategies in animal models is a prerequisite for any further translational development. In addition, laser-induced retinal damage in mice is a well-documented and widely used animal model. It reflects at least several aspects of the inflammatory and vascular component of human age-related macular degeneration. In detail, Bruch's membrane is being ruptured by laser coagulation leading to induced expression of vascular endothelial growth factor, uncontrolled vessel choroidal vessel growth into the retina and vascular leakage. Although the pathomechanisms for rupture of Bruch's membrane are of different nature, expression of VEGF, vessel ingrowth in murine models and leakage reflect key clinical pathomechanisms of wet AMD aside from inflammatory mechanisms. Rodent Laser CNV models are routinely used in pharmacological research and rodent-specific neutralization of VEGF-activity inhibits choroidal vessel growth which in analogy to the standard of care in human AMD.

Referee #2 (Remarks):

In this study Karlstetter et al explore the role of Siglec11 in modulating mononuclear cell activation *in vitro* and *in vivo*, using laser coagulation as a model of AMD. The authors nicely show that Siglec11, along with some of its ligands, are expressed in human retinas. I particularly liked their *in vitro* experiments using the THP-1 monocyte cell-line; the authors show strong evidence that Siglec11-ligand interactions suppress secretion of pro-inflammatory molecules and reactive oxygen species, and suppress phagocytic function in these cells. It is also shown that intravitreal injection of poly-sialic acid (polySia avDP20) robustly reduced mononuclear cell infiltration and activation, as well as vascular abnormalities following laser photocoagulation, which could potentially be of therapeutic benefit for diseases such as AMD, where chronic inflammation may occur. The authors claim these *in vivo* effects are mediated by interactions of polySia avDP20 with siglec11. They use a novel humanized mouse, expressing human siglec11, to elucidate the function of this molecule *in vivo*. The study is nicely designed and carefully executed, but a major weakness is that the *in vivo* data simply does not support a role for Siglec11 in any of the outcomes measured. Intravitreal injection of polySia avDP20 has a comparable effect on all outcomes measured in WT and Siglec11 mice.

*We agree with the reviewer that the data shown in the original manuscript did not allow to fully claim a direct role of SIGLEC11 in the animal model. We did not properly mark all relevant statistical significant outcomes and unintentionally selected wrong statistical comparisons for illustration in Figures 1D and 2B of the original manuscript. However, these data already demonstrated at first submission of the manuscript a significant effect of polySia avDP20 (0.2 mg) in SIGLEC11 transgenic mice vs. WT mice. Nevertheless, we took these concerns of the reviewer

very serious and repeated the experiments to increase the overall group size and provide more relevant exemplary images in Figure 1A, 1C, and 2A. These new results (discussed below as responses to the individual major comments) in our opinion clearly support specific therapy effects in the 0.2 mg polySia avDP20 treated SIGLEC11 mice that are not present in wild type mice.*

Major comments

1. The authors claim that polySia avDP20 has an anti-inflammatory effect and reduces vascular leakage in a Siglec11 dependent manner. The data they show do not support this:

- Fig 1 B: polySia avDP20 leads to an identical reduction in Iba1 stain in both WT and Siglec11 tg mice. The effect of polySia avDP20 is in no way different in WT vs Siglec11 tg mice.

*Where appropriate, we have added new exemplary images of microglia in Figure 1A and markedly increased the sample size in Figure 1B for the comparison of 0.2 µg polySia avDP20 treatment in SIGLEC11 transgenic versus wild type animals. The direct comparison of both groups now clearly shows that SIGLEC11 transgenic mice treated with 0.2 µg polySia avDP20 show significantly (**p=0.0046) reduced reactive phagocyte accumulation at the RPE/choroid when directly compared to treated WT mice (Fig 1B).*

- Fig 1D: There is a higher number of microglia per laser spot in the retina of WT mice when compared to the Siglec11 Tg mice after injection of the lower dose of polySia avDP20 (0.2ug). However, Siglec11 Tg mice have much lower levels of microglia per laser spot to begin with (when compared to WT after PBS injection). There does not seem to be a real difference in the effect of 0.2ug of polySia avDP20 vs PBS between the two genotypes. Additionally, when looking at the higher dose of polySia avDP20 (3ug), the effect of intravitreal injection on the number of microglia seems identical, with the exception of one (very large) outlier in the WT group.

*We provide new exemplary images of reactive retinal microglia in Figure 1C and increased the sample size in Figure 1D. The number of laser spots with activated microglia tends to be smaller in SIGLEC11 transgenic mice, potentially due to intra-retinal sialic acids that may activate SIGLEC11 signaling, but the percentage of laser spots with activated microglia is significantly lower in SIGLEC11 transgenic mice treated with 0.2 µg polySia avDP20 compared to wild type mice (Fig. 1D; ***p=0.0009).*

- Fig 2B: similar to the previous data, the high dose of polySia avDP20 does not seem all that different between WT and Siglec11 Tg mice. In contrast to data in figure 1, here the lower dose of 0.2ug seems to have a bigger effect in the Siglec11 Tg mice vs WT. If Siglec11 were to mediate the effects of polySia avDP20 one would expect much larger differences between WT and transgenic mice. If anything, data from figure 2B suggests that Siglec11 may contribute to the effects of polySia avDP20 on vascular leakage, but in no way do these data support Siglec11 as the main mediator of the in vivo effects.

*We thank the reviewer for re-addressing this issue on SIGLEC11 specificity also related to vascular leakage. As mentioned above for microglia activation, we also repeated, validated, and expanded our *in vivo* angiography experiments. These additional data clearly showed that SIGLEC11 transgenic mice treated with 0.2 mg polySia avDP20 had reduced vascular leakage compared to treated WT mice (Fig 2B; ***p<0.0001).*

2. Figure 3. In this paper the authors explore the function of Siglec11, for which there is no mouse homologue. However, in the experiments presented in EV Fig 3, instead of using microglia from their transgenic mouse expressing Siglec11, the authors knock-out SiglecE (or Siglec 9). While Siglec9 and 11 belong to the same family they are different molecules. In addition, humans express Siglec9. It is not clear to me why the authors studied SiglecE/Siglec9 in these in vitro experiments, but not in vivo. Are the effects they are seeing in both WT and Siglec11 Tg mice mediated by Siglec9? Why have the authors not looked at Siglec9 in their in vivo studies? The role of this Siglec in their study should at the very least be discussed.

*SIGLEC11 and SIGLEC9 are human Siglecs, while SiglecE is present in the mouse. Siglecs typically recognize a specific combination and linkage of 3-4 monosaccharides. SIGLEC11 binds to sialic acid (Neu5Ac) α2.8-linked to another sialic acid (Neu5Ac) that is further linked to another

sialic acid or galactose (Macauley et al. 2014, Nat Rev Immunol. Oct;14 (10):653-66). There is no direct homologue of SIGLEC11 in the mouse. However, Siglec-E, as putative closest murine homologue of SIGLEC9, has a relative broad binding spectrum to $\alpha 2.3$ -, $\alpha 2.6$ - and $\alpha 2.8$ -linked sialic acid. Recently, it has been shown that SiglecE binds to sialic acid (Neu5Ac) $\alpha 2.8$ -linked to another sialic acid (Neu5Ac) on nanoparticles (Spence et al. 2015, Sci Transl Med. Sep 2;7(303):303ra140.). Furthermore, Siglec-E like SIGLEC11 is expressed on microglia (Claude et al. 2013, J Neurosci. 2013 Nov 13;33(46):18270-6). In contrast, binding preference of SIGLEC9 is more specific and restricted to sialic acid (Neu5Ac) $\alpha 2.3$ -linked to galactose and also requires a sulfate residue in the glycan (Macauley et al. 2014, Nat Rev Immunol. Oct;14 (10):653-66).

We recently showed that polySia avDP20 (polymer of sialic acid/ Neu5Ac $\alpha 2.8$ -linked to sialic acid/ Neu5Ac) acts via SIGLEC11 at nanomolar concentrations (Shahraz et al. 2015, Sci Rep. 2015 Nov 19;5:16800). Here, in this study we demonstrate that polySia avDP20 also shows bioactivity via SiglecE, but at an approximately 10x higher concentration (low micromolar concentrations) compared to its bioactivity via SIGLEC11. We agree with the reviewer that a follow-up analysis of polySia avDP20 application at approximately 10x higher dose in SiglecE knock out mice compared to WT controls would be of particular interest, but we feel that this is out of the scope of the current manuscript.

Furthermore, we performed additional experiments to determine whether SIGLEC9 is expressed on THP1 macrophages that showed anti-inflammatory response to polySia avDP20. No SIGLEC9 was expressed on THP1 macrophages as determined by flow cytometry (please see Figure below).

Figure legend. No SIGLEC9 expression on THP1 macrophages. Investigation of SIGLEC11 and -9 presence on THP1 macrophages with human blood derived granulocytes serving as positive control. SIGLEC9 was detected on granulocytes but was absent on THP1 cells. SIGLEC11 was present on the cell surface of THP1 cells, but was undetectable on the granulocytes. Representative images of at least three independent experiments are shown. **Method flow cytometry based analysis of SIGLEC9 and -11 in vitro.** A novel monoclonal SIGLEC11 antibody (clone # 3EH, Abmart) and a monoclonal SIGLEC9 antibody (clone # 191240, R&D Systems) were used to stain THP1 macrophages and human peripheral blood granulocytes. Primary granulocytes were isolated from human blood samples using the Vacutainer CPT kit (BD, Ref: 362782). Cells were stained for protein expression with the primary antibodies for 1 hour (SIGLEC9 and -11 antibody dil. 1:500), followed by a PE fluorescence-labelled secondary antibody for 30 minutes (Jackson Laboratories Inc, USA, dil. 1:200). Control samples were incubated with an isotype control and secondary fluorescence labelled antibodies. Analysis was performed with a flow cytometer (BD, Calibur) and the FlowJo 8.7 Software (Tree Star Inc.).

3. EV Fig 1, panel F. The two halves of panel F very clearly do not belong to the same section/stain.

We apologize for this mix-up and now present a correct image in EV Fig 1 panel F.

4. EV Fig 2B. Here the authors perform flow cytometric analysis of brain tissue (which is never mentioned in the main text). It is not clear to me why the authors are examining expression of Siglec11 in the brain and why this is included with retina data.

We are sorry for this error in the figure legend. Data in EV Fig 2B are from the retina as pointed out in the headline of the figure. We now corrected the error in the description of the figure legend 2B. Data shown are from the mouse retina, not from the brain. We also performed flow cytometry analysis of the brain, but we decided to omit this analysis and replace it by the retina during preparation of the original manuscript.

Minor comments

1. Page 3. Reference 'FritscheIgl et a,;' needs to be corrected.

Corrected.

2. Page 5 and EV Fig.2A. The authors claim comparable levels of Siglec11 gene transcripts are detected in the retina of humans and transgenic mice. The gel shown in EV figure 2 appears to show lower levels of expression in the human retinas compared to mouse. The authors should add a graph with the normalized quantification of Siglec11 transcript levels.

The reviewer is correct in acknowledging a slightly lower expression in the depicted human retinal samples compared to the transgenic mice. We have therefore repeated the comparative analysis of SIGLEC11 transcription in 3 different WT mouse retinas, 3 different SIGLEC11 transgenic mouse retinas as well as 3 different human retinas (see revised Expanded View Figure 2A). Based on these findings and the data presented in Expanded View Figure 1A we now take note of the variability of SIGLEC11 RNA levels in retinas of different human donors and deleted the assumption "comparable levels".

3. Page 9. "as well as the protein release of VEGFA (Fig. 6D)." Do the authors mean Figure 3B?

Corrected. Now figure 3B.

4. Figure 3 is hard to read, please add labels (e.g. B, D) to the two top right panels (Tnfa protein and VEGFA protein)

Corrected. We added labels (B, D) to the two top right panels.

5. The authors should consider changing the graphs in Figure 3 to compare treatments and not genotype (as is). In the text the authors mostly describe differences between treatment in WT THPs, which are no longer observed in the THP1 Siglec11/16 KO, rather than directly compare e.g. the

effect of LPS on WT vs KO THP cells. It would make it easier for the reader to follow the text if the graphs were changed accordingly.

We tested another version of the graph to compare treatments, which was very busy due to 5 different treatment schemes. We feel that this other version does not make it easier to the reader to find the relevant information. Therefore, we kept the original setup of the graph in figure 3. However, we realized that the SIGLEC11/16 KO THP1 cells were poorly described in our original manuscript. Therefore, we now added more details describing the generation and validation of SIGLEC11/16 KO THP1 cells (please see supplementary methods and supplementary figure 3).

2nd Editorial Decision

31 October 2016

Thank you for the submission of your revised manuscript to EMBO Molecular Medicine. Please accept my sincere apologies for the time it took to get back to you.

You will see that while referee 2 was now fully supportive, referee 1 was not. As we felt that these issues were important enough to affect significance of the work if relevant, we asked a Siglec expert for external advice. We have now received it and the adviser is fully supportive and enthusiastic about the data performed. I am pleased to inform you that we will be able to accept your manuscript pending the following final amendments:

1) Please address in writing the remaining concerns of referee 1. Please provide a letter INCLUDING the reviewer's reports and your detailed responses to their comments (as Word file).

Please submit your revised manuscript within two weeks. I look forward to seeing a revised form of your manuscript as soon as possible.

***** Reviewer's comments *****

Referee #1 (Comments on Novelty/Model System):

I have reviewed the paper once more with regard to the use of the Iba-1 promoter and as indicated in my comments, the targeting of Siglec 11 to this promoter may not be sufficient to definitively exclude bone marrow derived cells from the microglial cell population

Referee #1 (Remarks):

The authors do not convincingly present the case that the effects which they see in this model system are specific for Siglec 11 or E or even Siglecs in general. Is it possible to perform some specific knock down experiments which would demonstrate specificity? Also while yolk sac-derived resident microglia express IBA-1, bone marrow derived perivascular macrophages also can express Iba-1 which undermines the specificity of the transgenic mouse model used here (see reference: Greter M, Lelios I, Croxford AL. Microglia Versus Myeloid Cell Nomenclature during Brain Inflammation. *Frontiers in immunology*. 2015;6:249.)

Referee #2 (Remarks):

The authors have adequately responded to and addressed the comments from my critique.

2nd Revision - authors' response

15 November 2016

Referee #1 (Comments on Novelty/Model System):

I have reviewed the paper once more with regard to the use of the Iba-1 promoter and as indicated in my comments, the targeting of Siglec 11 to this promoter may not be sufficient to definitively exclude bone marrow derived cells from the microglial cell population.

*We agree with the reviewer that targeting SIGLEC11 by the Iba-1 promoter also leads to expression of SIGLEC11 on bone marrow derived tissue macrophages. Indeed, we show in Supplementary Figure 1 and in Supplementary Table 1 that *SIGLEC11* gene transcripts are also present in spleen and liver.

We therefore believe that the expression of SIGLEC11 in the humanized SIGLEC11 transgenic animal models used in this study very well reflects the human situation with low expression levels of SIGLEC11 on microglia and tissue macrophages. We also would like to point out that the novelty of this study is the protective effect of low molecular weight polysialic acid on retinal damage. The transgenic animal model is one of the applied tools to obtain data with higher relevance for human diseases.

Referee #1 (Remarks):

The authors do not convincingly present the case that the effects which they see in this model system are specific for Siglec 11 or E or even Siglecs in general. Is it possible to perform some specific knock down experiments which would demonstrate specificity? Also while yolk sac-derived resident microglia express IBA-1, bone marrow derived perivascular macrophages also can express Iba-1 which undermines the specificity of the transgenic mouse model used here (see reference: Greter M, Lelios I, Croxford AL. Microglia Versus Myeloid Cell Nomenclature during Brain Inflammation. *Frontiers in immunology*. 2015;6:249.)

*We don't agree with the reviewer that the effects of polySia, which we see in this model system, are not specific for SIGLEC11. We show that low dose polySia (0.2 mg) prevents macrophage activation/recruitment in the humanized SIGLEC11-transgenic mice, but not in the littermate control mice (please see Figure 1A and B). Furthermore, we show that low dose polySia (0.2 mg) prevents microglial activation in the humanized SIGLEC11-transgenic mice, but not in the littermate control mice (please see Figure 1C and D). Finally, we show that low dose polySia (0.2 mg) prevents vascular leakage in the humanized SIGLEC11-transgenic mice, but not in the littermate control mice (please see Figure 2A and B).

Although we have strong evidence that polySia acts at a low dose dependent on SIGLEC11 in this animal model, we also show that higher concentrations of polySia can act on the complement system (Figure 2C and D; Figure 3H). Thus, polySia acts synergistically on the innate immune system by preventing oxidative damage and complement activation.

We are aware that Iba-1 genes and proteins are transcribed and expressed on microglia, perivascular macrophages and other tissue macrophages. As mentioned above (Novelty/Model system) we have not claimed that the SIGLEC11 transgenic mice driven by the Iba-1 promoter show exclusively microglial-restricted expression.

Referee #2 (Remarks):

The authors have adequately responded to and addressed the comments from my critique.

3rd Editorial Decision

21 November 2016

Thank you for the submission of your revised manuscript to EMBO Molecular Medicine.

Unfortunately, the manuscript format is not fully adequate for the paper to be accepted yet. Please address ASAP the following [final editorial requests]. Please submit your revised manuscript as soon as possible.

Authors made requested changes.

Corresponding Author Name: Harald Neumann and Thomas Langmann

Manuscript Number: EMM-2016-06627